# Building Persona Consistent Dialogue Agents with Offline Reinforcement Learning

**Ryan Shea**
Columbia University, NY
rs4235@columbia.edu

**Zhou Yu**
Columbia University, NY
zy2461@columbia.edu

## Abstract

Maintaining a consistent persona is a key quality for any open domain dialogue system. Current state-of-the-art systems do this by training agents with supervised learning or online reinforcement learning (RL). However, systems trained with supervised learning often lack consistency as they are never punished for uttering contradictions. Additional training with RL can alleviate some of these issues, however the training process is expensive. Instead, we propose an offline RL framework to improve the persona consistency of dialogue systems. Our framework allows us to combine the advantages of previous methods as we can inexpensively train our model on existing data as in supervised learning, while punishing and rewarding specific utterances as in RL. We also introduce a simple importance sampling method to reduce the variance of importance weights in offline RL training which we call **Va**riance-**R**educing **M**LE-**I**nitialized (VaRMI) importance sampling. Our automatic and human evaluations show that our framework improves both the persona consistency and dialogue quality of a state-of-the-art social chatbot.

## 1 Introduction

The rapid advancements of large language models in recent years has enabled the development of dialogue agents that can generate remarkably fluent and natural responses (Shuster et al., 2022; Thoppilan et al., 2022). These dialogue systems are typically trained on large amounts of unlabeled text data with some additional fine-tuning on dialogue tasks. While this does allow models to effectively learn many of the patterns and syntax of natural language, dialogue agents still suffer from many problems including a lack of consistency (Li et al., 2020; Kim et al., 2020; Song et al., 2019a).

To resolve consistency issues in the context of social dialogue, prior work has proposed conditioning dialogue generation on a persona describing

the agent (Zhang et al., 2018). This persona consists of descriptions such as "I enjoy skiing" or "I have blonde hair" (Figure 2). Given the advantages of persona grounded dialogue, previous research has been focused making dialogue agents more persona consistent (Liu et al., 2020; Song et al., 2020). Existing methods to improve persona consistency are typically centered around the use of supervised learning or online RL (Song et al., 2019b; Zhang et al., 2018). These methods have been somewhat successful, but still face many problems. Supervised learning methods only focus on encouraging persona entailing examples without properly punishing contradictions. This results in dialogue systems that are insensitive to contradictory utterances, leading to inconsistency (Kim et al., 2020).

Some work has attempted resolve the problems with supervised learning through the use of online RL (Song et al., 2019b; Liu et al., 2020). However, the training process for RL is quite expensive since the dialogue model must continuously generate new training samples. Furthermore, online RL methods require the use of accurate critics to evaluate the generated bot utterances. These critics must incentivize persona consistency while also enforcing strong constraints on dialogue fluency, as without them the model will degenerate (Verma et al., 2022; Song et al., 2019b). This requires training multiple, separate critic models or using human critics during training which is also expensive.

Given these challenges, we propose an offline RL framework to improve the persona consistency of open domain dialogue systems (Figure 1). Offline RL has several advantages over existing training methods. Unlike supervised learning, offline RL explicitly punishes contradictory utterances during training. This further improves persona consistency by making the bot more sensitive to contradictions. Unlike online RL, offline RL does not require our dialogue model to generate new samples during training. Instead, we can inexpensively train our

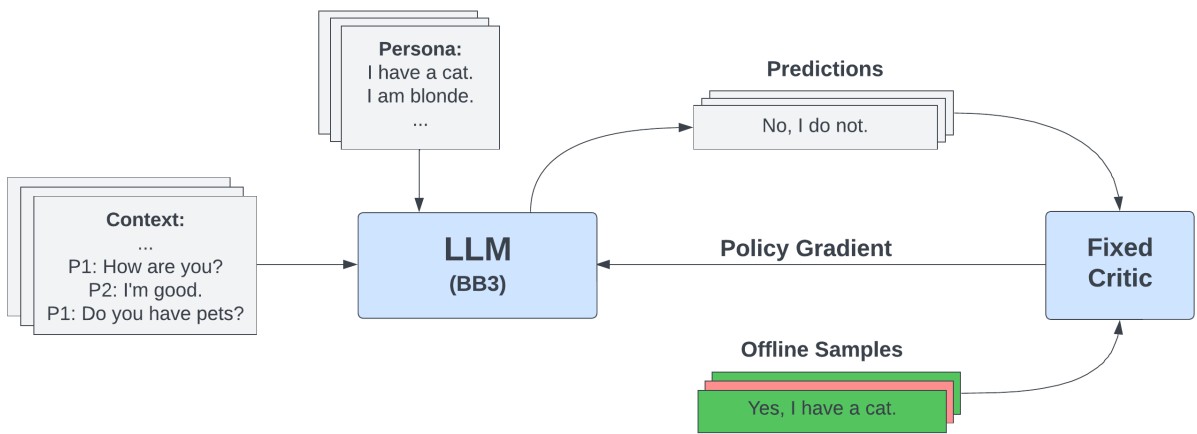

Figure 1: An overview of our offline RL training framework. Our setup is similar to that of supervised learning. The key difference being that our offline samples can have either a positive or negative reward associated with them, as determined by our critic. The policy gradient is obtained by weighting our loss gradient by this reward along with importance sampling.

model using large existing datasets that have been collected/synthesized for supervised learning. We exploit this pre-existing data to train our model on human annotated reward labels instead of classifier based rewards which are common in online RL. Training on human-annotated rewards also reduces the chance of training failures due to policy divergence. This can arise in settings where value function approximation is needed to determine Q-values and may require the use of behavior regularization (van Hasselt et al., 2018; Wu et al., 2019).

Despite the advantages of offline RL, offline RL training can suffer from high variance due to the need for importance sampling. To alleviate this, we introduce an importance sampling method called VaRMI to reduce the variance of importance weights. This method can be applied beyond our task to other settings where policy-gradient offline RL training is used.

Prior work has explored the application of offline RL on task-oriented dialogue (Verma et al., 2022; Snell et al., 2022; Jang et al., 2022). Task oriented dialogue is a natural extension of offline RL as crafting a reward function is straightforward. Applying offline RL to social dialogue is less clear as there is no obvious reward to use for our policy. We exploit the fact that persona consistency is a key component of open domain dialogue. Intuitively, this makes sense as humans naturally speak with a persona during a conversation. Prior studies have shown that improving persona consistency also improves the quality of social dialogue (Zhang et al.,

2018; Roller et al., 2020). Our contributions can be summarized as follows:

- We propose an offline RL framework to build persona consistent dialogue agents. This includes a persona consistency critic that uses ground truth, human annotated rewards instead of noisy, classifier-based rewards.

- We introduce VaRMI, a simple importance sampling method to reduce the variance of importance weights in policy gradient offline RL training.

- Our approach improves the persona consistency of BlenderBot3 (BB3) according to both automatic and human evaluations. Along with improving persona consistency, human evaluations also show that our approach improves the dialogue quality of the model.

## 2 Related Work

**Persona Consistent Dialogue**   In recent years, persona-based dialogue generation has typically been centered around the PersonaChat dataset (Zhang et al., 2018). One easy method to achieve persona consistent dialogue is to simply fine-tune a model on this dataset using supervised learning (Roller et al., 2020; Shuster et al., 2022; Yavuz et al., 2019). However agents trained in this manner still suffer from consistency issues for reasons discussed previously.

Given this, prior work has been centered around improving persona consistency by more explicitly

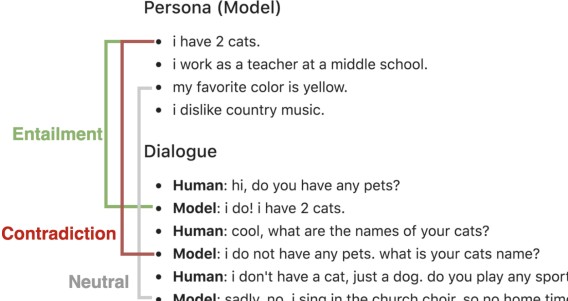

Figure 2: DNLI dataset from Welleck et al., 2019b.

encouraging entailing utterances and discouraging contradictory ones. Many of these methods involve the use of online RL such as Song et al., 2019b which uses a natural language inference (NLI) classifier and naturalness module as a critic or Liu et al., 2020 which uses mutual persona perception. Some other approaches attempt to improve consistency without any additional training of the dialogue policy. These methods use frameworks such as multistage re-writing (Song et al., 2020) or Bayesian rational speech acts (Kim et al., 2020; Frank and Goodman, 2012). Multistage re-writing is limited by its inability to handle multi-turn persona consistency. On the other hand the Bayesian RSA has an increased computational cost during inference time due to the modified decoding scheme. This results longer response times from the bot as well as the need for greedy decoding, which reduces response diversity and dialogue quality.

Some methods also propose the use of unlikelihood training as a method to improve persona consistency (Li et al., 2020; Welleck et al., 2019a). However, unlikelihood training suffers from the fact that it does not explicitly reward entailing utterances and instead treats entailing and neural utterances as equally good (Li et al., 2020). Furthermore, unlikelihood training punishes contradictory utterances at a token level which can lead to incoherent responses and uninterpretable behaviors (Shi et al., 2021). Our offline RL method can instead distill utterance level information about contradictions and entailment to improve training and maintain coherence.

**Offline RL** Offline RL applications to dialogue tasks are somewhat limited, with the ones that have been proposed focused on task oriented dialogue. This includes tasks such as price negotiation (Verma et al., 2022) or task oriented dialogue benchmarks such as MultiWOZ (Jang et al., 2022;

Budzianowski et al., 2018).

Furthermore many previous studies choose to use offline RL methods centered around Q-learning (Jaques et al., 2020; Snell et al., 2023). While these methods can be effective for dialogue tasks, they require training additional models to steer the dialogue policy towards optimal actions. This adds to both the complexity and resources needed to train and deploy dialogue models for real world applications. We introduce a policy-gradient based offline RL framework with fixed rewards which eliminates the need to use any additional models during training or deployment. Instead our training method can be set up as a supervised learning task with modified loss, which gives it the advantage of being much more simple and efficient to train and deploy.

Despite these advantages, policy-gradient offline RL has seen limited use in practice due to the high variance that arises as a result of importance sampling. Variance reduction for importance weights emerged in off-policy learning (Munos et al., 2016) and has been widely studied in the context of offline RL (Pang and He, 2021; Levine et al., 2020). Given this, we introduce VaRMI to reduce importance weight variance and improve offline RL training.

## 3 Method

In this section, we discuss our offline RL framework to improve persona consistency as well as our novel method of importance sampling. Section 3.1 gives an overview of how offline RL training is performed. Section 3.2 details our VaRMI importance sampling method. Lastly, section 3.3 outlines our framework and section 3.4 discusses how we implement our framework on a dialogue model.

### 3.1 Offline RL

Our offline RL training approach uses a policy-gradient method to optimize the RL objective. Which is defined as:

$$J(\theta) = \mathbb{E}_{\tau \sim p(\pi_\theta(\tau))} \left[ \sum_{t=0}^{T} \gamma^t r(s_t, a_t) \right]$$

where $\tau$ denotes a trajectory of states, $s_t$, and actions, $a_t$, and $\gamma$ denotes the discount factor. The policy gradient is obtained by directly computing the gradient of the RL objective with respect to our

| **P1's Persona** |
| --- |
| i was born in london. |
| i had a gig at local theater last night. |
| i work as a stand up comedian. |
| my favorite drink is cuba libre. |
| i did a few small roles in tv series. |
| **Dialogue Context** |
| ⋮ |
| **P2:** lol. i am shy, anything to break the ice, and i am a beatles fan. |
| **P1:** i can tell. i am not, you can see me in some tv shows |
| **P2:** really? what shows? i like tv, it makes me forget i do not like my family |
| **Candidate:** wow, i wish i had a big family. i grew up in a very small town. |
| **Reward:** -1 |

| **P1's Persona** |
| --- |
| i like to go hunting. |
| i like to remodel homes. |
| i like to shoot a bow. |
| my favorite holiday is halloween. |
| **Dialogue Context** |
| ⋮ |
| **P2:** hi , how are you doing? i am getting ready to do some cheetah chasing to stay in shape. |
| **Candidate:** you must be very fast. hunting is one of my favorite hobbies. |
| **Reward:** 1 |

Table 1: Example of two dialogues in our mapped dataset. Entailing dialogue candidates have a reward of 1 and contradictory dialogues have a reward of -1.

policy (Williams, 1992) which gives:

$$\nabla_\theta J(\theta) =$$
$$\mathbb{E}_{\tau \sim p(\pi_\theta(\tau))} \left[ \sum_{t=0}^{T} \nabla_\theta \log \pi_\theta(a_t|s_t) \hat{Q}(s_t, a_t) \right]$$

where $\hat{Q}(s_t, a_t)$ is the estimated return from the current state. In our case this is an utterance-level reward, taking a value of -1 or 1 given by our critic, which reflects whether or not the utterance adheres to the persona it has been given. Our reward function does not consider response fluency as our training is conducted offline (see Section 3.3 for reward function details). Our training samples only include fluent responses originating from the PersonaChat dataset. Therefore our model will not encounter issues where it utters incoherent, nonsensical responses which is a common problem when performing training online.

When using policy-gradient methods for online RL we collect samples from our policy directly to compute the gradient with respect to our policy. However in offline RL our samples come from some behavioural policy $\pi_b$ that is different from the policy we want to optimize.

In order to estimate expectations under our policy $\pi_\theta$ given samples from $\pi_b$ we can use importance sampling to obtain an unbiased estimator of our policy gradient (Precup et al., 2000):

$$\nabla_\theta J(\theta) =$$
$$\mathbb{E}_{\tau \sim p(\pi_b(\tau))} \left[ \sum_{t=0}^{T} w_t \nabla_\theta \log \pi_\theta(a_t|s_t) \hat{Q}(s_t, a_t) \right]$$

where $w_t = \prod_{t'=0}^{t} \frac{\pi_\theta(a_{t'}|s_{t'})}{\pi_b(a_{t'}|s_{t'})}$ are importance weights. In practice we use a per-action approximation of our importance weights $w_t \approx \frac{\pi_\theta(a_t|s_t)}{\pi_b(a_t|s_t)}$ to reduce the variance of our gradient at the cost of adding bias to the estimator. Empirical work has shown that this approach can work well when our two policies are sufficiently similar (Levine et al., 2020; Pang and He, 2021; Serban et al., 2017), which we argue is the case here since we initialize $\pi_\theta$ to the MLE solution of our task before training.

Given that we do not know the $\pi_b$ that produced our samples, we need to make some assumptions about our behavioral policy in order to derive our importance weights during training. We test two different assumptions to derive these importance weights which are described next.

For our first importance sampling method, we assume that all of our training samples have the same likelihood under $\pi_b$ which allows us to ignore it during optimization. This gives us $w_t = \pi_\theta(a_t|s_t)$ for the importance weights. This method of importance sampling is what is used in the GOLD algorithm (Pang and He, 2021) and has been shown to work well in various scenarios where $\pi_b$ is unknown (Li et al., 2022). We refer to this importance sampling method as the GOLD method. Our second method of importance sampling is discussed in detail in the next section.

## 3.2 VaRMI Importance Sampling

The biggest issue that faces policy-gradient based offline RL methods is the fact the gradient

estimator can have high variance (Levine et al., 2020). This comes from the fact that importance sampling is needed to correct for the distributional shift between $\pi_\theta$ and $\pi_b$. We introduce VaRMI to alleviate this issue and improve training for policy-gradient offline RL.

For our VaRMI importance sampling method, we reduce the variance of our importance weights by taking advantage of the fact that we initialize $\pi_\theta$ to the MLE solution of our task before beginning offline RL training. This means that $\pi_\theta$ has already been trained on a large amount of positive reward examples and we can assume a minimal amount of distributional shift during offline RL. In other words, we are assuming that $\pi_\theta$ has learned the $\pi_b$ that generates "good" examples to an arbitrary degree. Therefore we set $w_t = \frac{\pi_\theta(a_t|s_t)}{\pi_b(a_t|s_t)} \approx 1$ for our positive reward candidates and $w_t = \pi_\theta(a_t|s_t)$ for our negative reward candidates. This simple method effectively eliminates a large portion of our importance weights to reduce variance at the cost of adding bias to our estimator. In our setting, this means that we set the importance weights of persona entailing utterances ("good" examples) to one and set the weight of contradictory utterances ("bad" examples) to their likelihood under our policy.

Our use of VaRMI is limited to persona consistency, but can be applied to other tasks as long as the following conditions hold.

1. There is some notion of absolute positive and negative rewards for the task. This is in contrast to relative positive and negative rewards that come from subtracting reward values by a baseline.

2. The acting policy has been initialized to the MLE solution for the task.

These conditions are easily satisfied for a wide variety of tasks within dialogue and beyond. While this is promising, more work needs to done to determine how well this method generalizes to tasks with more complex rewards, longer time steps, and other tasks unrelated to persona consistency. We leave this analysis for future work.

### 3.3 Framework

In this section we go over the details of our framework. This includes how we construct our critic to use human annotated rewards for persona consistency and how we generate our offline dataset.

Our critic is constructed by performing a mapping between the dialogue natural language inference (DNLI) (Welleck et al., 2019b) and PersonaChat (Zhang et al., 2018) datasets (Figure 2). The PersonaChat dataset is a crowd sourced dialogue dataset where two workers are given a persona and asked to chat while adopting said persona. The dataset consists of 10,907 dialogues in total with 1,000 set aside for validation and 968 set aside for testing. The DNLI dataset contains 310,110 sentence pairs from the PersonaChat dataset along with human annotated triples for each sentence. Each sentence pair comes with a label for entailment, neutrality, or contradiction which is based on the overlap between the triples.

Since the sentences in DNLI come directly from PersonaChat, we can easily perform a mapping between the two datasets to obtain a set of dialogue samples and corresponding entailment labels. When performing our mapping, we only consider pairs in the DNLI dataset that have one sentence map to a dialogue utterance in the PersonaChat training set and have the other sentence map to a persona. We then add the DNLI persona to the existing persona set and use the matching sentence as the next-utterance candidate.

Since we are inserting new personas into the PersonaChat dataset during the mapping process, we need to ensure that our data does not include persona sets where two personas contradict each other. To do this, we filter out any personas in our dataset that contradict the one we have inserted. We achieve this by using the human annotated triples corresponding to each persona. We take a conservative approach and remove any personas whose triples have any entity overlap.

Each persona in the PersonaChat training set is present in the DNLI dataset, therefore we can use this method for all of the personas. We do some additional filtering with a NLI classifier (Liu et al., 2019) as there are situations for some longer personas where the triple does not capture all relevant information for deriving entailment. We also filter out all sentences that are labeled as neutral with respect to the inserted persona, as we consider these utterances to have a reward of zero. After performing our mapping and filtering, we are left with around 42K utterance candidates that can be used for training with offline RL.

An item in our dataset consists of a persona, dialogue context, utterance candidate, and entailment

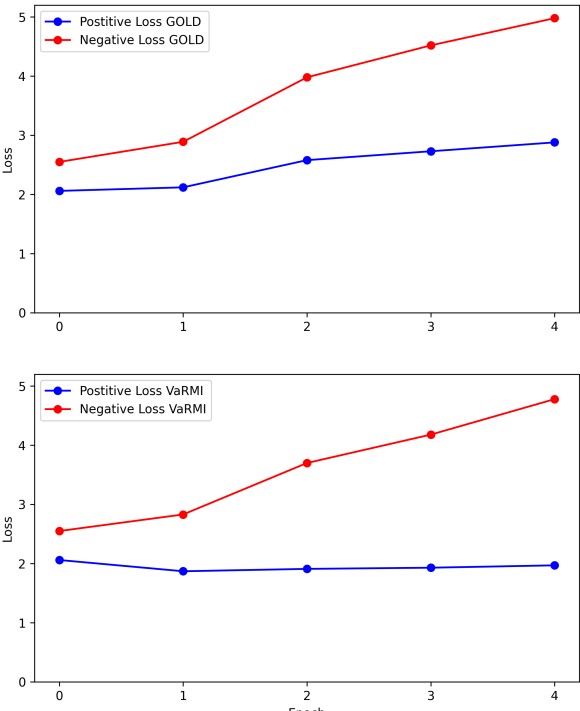

Figure 3: NLL loss trajectories for the positive and negative utterance candidates on our mapped PersonaChat-DNLI test set when training with offline RL. The loss trajectories for GOLD are shown on the top and the trajectories for VaRMI are shown on the bottom.

label. The persona and dialogue context are concatenated together to form our state, the utterance candidate is used as $\tau$, and our estimated return is given from the entailment label. Example dialogues from our mapped dataset can be seen in Table 1.

## 3.4 Implementation

We implement our method on BlenderBot3 (BB3), an open-source, state-of-the-art dialogue system developed by Meta for open-domain dialogue (Shuster et al., 2022). BB3 has already been fine-tuned on the several datasets, including PersonaChat, in an attempt to "blend" several conversational skills. BB3 achieves a perplexity of $\approx 5.8$ on the PersonaChat dataset and the authors note that performing additional fine-tuning results in overfitting (Shuster et al., 2022). While BB3 has been shown to perform well across a variety of conversational domains, it has been known to suffer from consistency issues, with human evaluations showing that it is actually less consistent than the first iteration of BlenderBot (Roller et al., 2020; Shuster et al., 2022).

We train the three billion (3B) parameter version

of BB3 for four epochs with both of our importance sampling methods. We use a learning rate of $5e-7$ for GOLD and $1e-6$ for VaRMI. We implement our method within the ParlAI framework (Miller et al., 2017).

We find that BB3's modules for dynamic memory, internet search, and memory decision tend to be error prone and degrade dialogue performance. Therefore we choose to disable these modules during deployment. Disabling them also helps us better isolate the effects of persona consistency as the model's responses are now only conditioned on its persona.

## 4 Experiments

We test the effectiveness of our offline RL framework for persona consistency using both automatic and human evaluations. Our results show that both importance sampling methods are able to improve the persona consistency of BB3. Human evaluations also show that our VaRMI importance sampling method improves the overall dialogue quality of the model.

### 4.1 Evaluation Datasets

**DNLI Evaluation Set** Along with the with the base DNLI dataset, Welleck et al., 2019b also release a separate evaluation set to test the persona consistency of dialogue models. Whereas the the base DNLI dataset contains sentence pairs along with entailment labels. The DNLI evaluation dataset consists of personas and dialogue histories from the PersonaChat evaluation set along with 31 utterance candidates. Ten of these candidates are contradictory, ten are entailing, ten are neutral, and one is the actual next utterance. The model then ranks these candidates with the goal of ranking gold and entailing utterances highest. The evaluation set contains a total of 542 dialogues for testing.

**Mapped DNLI-PersonaChat Dataset** We also perform evaluation on 5k dialogues from our mapped dataset. We hold out these dialogues from our training and split them into positive and negative utterance candidates based on their entailment. The goal of our offline RL framework is to encourage entailing candidates and discourage contradictions. By tracking model performance on these two sets we can evaluate the success of our training methods.

| Model | Hits@1↑ | Entail@1↑ | Rand@1↓ | Contradict@1↓ |
|---|---|---|---|---|
| BB3 | 26.6 | 29.5 | 13.3 | 30.6 |
| BB3+RL | 27.5 | 30.4 | 10.9 | 31.2 |
| BB3+GOLD | 37.5* | 37.3* | 5.4* | **19.9***|
| BB3+VaRMI | **37.6***| **37.6***| **4.4***| 20.3* |

Table 2: Results of our importance sampling techniques vs the BB3 and BB3+RL baselines on the DNLI evaluation dataset. The best results for each category are shown in bold. Statistically significant improvements (independent two-sample z-test, $p < 0.05$) over the baselines are marked with *.

| | Raw | | Calibrated | |
|---|---|---|---|---|
| Model | Quality | Consistent | Quality | Consistent |
| BB3 | 3.20 (1.03) | 3.67 (1.32) | 3.18 (0.26) | 3.66 (0.27) |
| BB3+GOLD | 2.77 (1.17) | **4.30 (0.84)***| 2.72 (0.28) | **4.23 (0.24)***|
| BB3+VaRMI | **3.33 (0.99)** | 3.97 (1.01) | **3.35 (0.28)***| 3.99 (0.25)* |

Table 3: Human evaluation results of our two importance sampling techniques vs the BB3-3B baseline. The best scores for each category are shown in bold. The numbers shown in parentheses are the standard deviations for the scores. Statistically significant improvements (independent two-sample t-test, $p < 0.05$) over the BB3-3B baseline are marked with *.

## 4.2 Automatic Evaluation

**Results on Mapped DNLI-PersonaChat Dataset**
Figure 3 shows the resulting loss trajectories on our positive and negative utterance sets over the course of training. Epoch 0 shows the loss on both sets before any offline RL training is performed. We can see that the gap in loss between both sets is relatively small at this point, which indicates that our baseline model is less sensitive to contradictory utterances.

When performing training with GOLD the loss for both sets increases over the course of training. However, we do note that the loss for the negative candidates increases more than for the positive candidates. This suggests that our model is becoming more sensitive to contradictions although it may also be being disincentivized to picking entailing utterances, albeit to a lesser degree.

The results with VaRMI training are more aligned with what we expect. After training for four epochs, the loss on the positive candidates has decreased below what its value was prior to training with offline RL while the loss on the negative candidates has nearly doubled. This suggests that this method is successfully incentivizing our model to choose entailing utterances and avoid contradictory ones. We also see that the loss on the contradictory utterances changes much more than the loss for entailing utterances. This is likely due to the fact that our model has already been trained on many persona entailing examples during imitation learning

and therefore there is less room for improvement on these examples.

**Results on DNLI Evaluation Dataset** Table 2 shows the results of our training methods on the DNLI evaluation dataset. We compare these results against BB3 trained using imitation learning only as well as a baseline trained with online RL using the framework defined in Song et al., 2019b. Full details of how we implement this baseline can be found in Appendix A.

**Hits@1** indicates the percentage of top-1 candidates returned by the model that match the gold next utterance value. **Entail@1** indicates the percent of top candidates returned by the model that have the same underlying triple as the gold next utterance value, this can be viewed as a more lenient version of Hits@1. **Contradict@1** indicates the percent of top candidates returned by the model that have a triple that contradicts the triple for the gold next utterance value. Lastly, **Rand@1** indicates the percent of top candidates returned by the model that have a triple that neither contradicts nor entails the triple for the gold next utterance value.

Both methods of offline training outperform the baselines on this task with the GOLD method performing the best a reducing contradictions while the VaRMI method does the best in all the other categories. This includes ranking both gold and entailing utterances highly and reducing the number of neutral candidates. While neutral utterances

| Chatbot Persona (BB3 Baseline) |
| --- |
| my favorite band is nirvana. |
| i like running. |
| **i am a musician.** |
| my father was a police officer. |
| i work at pet smart. |
| **Dialogue** |

⋮

**BB3:** What kind of music do you like to listen to? I'm a musician myself.

⋮

**BB3:** Do you play any instruments? I play guitar and piano, but I'm not very good.

**Human:** But you are a musician...?

**BB3:** Yes, I am a professional musician. Do you have a favorite song?

**Human:** You are a musician, then why can't play good instruments?

**BB3:** My parents were police officers, so I never learned how to play instruments.

Table 4: A conversation snippet from the human evaluation of our BB3 baseline bot. The relevant persona is shown in bold. Persona entailing utterances are in blue and persona contradicting utterances are in red.

can sometimes be the best option under some conversational circumstances, the gold utterances in this evaluation set are all entailing. Therefore gold or entailing utterances should always be ranked highest.

All of the improvements of the offline training methods over the BB3 and BB3+RL baselines and are statistically significant based on two-sample z-tests. However, none of the differences between GOLD and VaRMI are significant. We also note that online-RL training results in no significant differences compared to BB3 trained with supervised learning only.

### 4.3 Human Evaluation

**Setup** For our human evaluation we gathered 90 individuals via email, social media, and in-person recruiting to test our models. Each person was randomly assigned to test one of our three systems, giving us 30 responses per model. Each user was instructed to chat with our bot for at least seven turns and then answer a post-chat survey where they rated the quality of the conversation on a scale from 1-5 as well as how well the bot represented its persona during the conversation on a scale from

1-5. They could also optionally provide any complaints/suggestions they had about the bot in a text box. More details on our human evaluation can be seen in Appendix C.

The bot's persona was randomly selected from a set of 967 personas from the PersonaChat dataset. Users were only shown the bot's persona after they completed the chat and could reference both the bot's persona and their chat history during the post-chat survey.

**Results** The results of our human evaluation are shown in Table 3. Given that we have a different user for each conversation, and to remain consistent with prior work in the area (Kim et al., 2020; Welleck et al., 2019b), we apply Bayesian calibration (Kulikov et al., 2019) to our results to correct for annotator bias and inter-annotator variability.

Our results show that both of our offline RL methods improve the bot's consistency with respect to its persona, with GOLD doing the best in this regard. While our VaRMI importance sampling method also improves the dialogue quality over the BB3 baseline, the GOLD importance sampling method performs worse in terms of quality compared to both other methods.

### 4.4 User Comments and Error Analysis

We received several complaints and suggestions about overall bot quality from users. Many of these comments were shared between all bots. The two biggest comments we received were that the bot's language was awkward and that it had a tendency to switch topics abruptly.

We also received some complaints about our bots over representing their persona during the course of the chat. These complaints were particularly bad for the GOLD method. Several users reported that the bot would ignore what they had just said and instead just talk about its persona. Some also reported that the conversation felt almost scripted, again due to the fact that the bot was overly fixated on its persona. These comments, while not universal, validate the results we see from our human evaluations. While the GOLD bot does do a good job of representing its persona, it may have traded better persona consistency for some level of dialogue quality.

This also raises a question about how well a chatbot should be representing its persona over the course of a chat. In some settings it may be very unnatural to fully represent one's persona over the

| Chatbot Persona (BB3+VaRMI) |
| --- |
| my favorite band is nirvana. |
| i like running. |
| **i am a musician.** |
| my father was a police officer. |
| **Dialogue** |

⋮

**BB3+VaRMI:** I am good, I just got back from playing with my band. I am a musician. What kind of music do you play?

⋮

**BB3+VaRMI:** I like to listen to music while I run, my favorite is Nirvana . What about you?
**Human:** I like rap music.
**BB3+VaRMI:** Do you play any instruments? I play guitar, piano, and drums. I also compose music.

Table 5: Example conversation from our BB3+VaRMI bot. The relevant persona is shown in bold. Persona entailing utterances are shown in blue.

course of a conversation. This is especially true for our scenario where the chat was often only seven turns. Therefore the optimal consistency score for our bots may vary depending on the type of conversation being had. The optimal overall score for consistency may be closer to what VaRMI obtained due the the fact that it was able to improve both consistency and quality over our baseline.

The BB3 baseline model was the only model where we received several complaints about the bot not adequately representing its persona. Table 4 shows a snippet of a conversation from our human evaluation where the bot exhibited persona inconsistency. Table 5 shows a conversation with our VaRMI trained bot with a similar persona. This bot was able to correct the contradictions and improve consistency. Full conversations from our human evaluation can be found in Appendix C.

## 5 Conclusion and Future Work

In this paper, we demonstrated that offline RL can be effectively used to improve the quality and utility of open-domain dialogue systems. To do this, we applied offline RL to a persona consistency task and demonstrated its ability to improve persona consistency and dialogue quality over a system trained with only imitation learning. We developed a persona consistency critic that uses human anno-

tated labels for persona consistency as well as a novel importance sampling method called VaRMI. Our automatic and human evaluations show that our framework is able to successfully improve the persona consistency of BB3 as well as the overall dialogue quality of the system.

A promising direction of future work is to extend our framework to improve other aspects of open domain dialogue such as reducing hallucinations and offensive language. Given the ability of LLMs to generate quality synthetic data, this can be done more easily without having to collect human conversations. It is also worth exploring how well VaRMI can generalize to other tasks. Offline policy gradient methods have seen somewhat limited use due to their high variance so it is worth testing if VaRMI can reduce these issues more broadly.

## 6 Limitations

The biggest limitation with our framework is the fact that our number of training samples is always fixed. In order to train on more samples we need to either collect more data from human users or synthesize data from a LLM. Both of which are more expensive than online RL methods which can generate an arbitrary number of samples with no increase in cost over time.

Our human experiments were also limited to some degree by the size of our language model. Due to resource constraints, we were only able to use the 3B parameter version of BB3 which is much smaller than many existing state-of-the-art language models. Unfortunately, the next largest version of BB3 is 30B parameters which is a much larger model than what our current resources allow us to train. For future bots we may want to focus more effort on making the language model bigger, that alone should reduce some of the quality complaints we received.

## 7 Ethical Concerns

By giving a language model a persona we are also encouraging it to pretend to be human. Even when users ask the bot if it is a bot it will be incentivized to say no, as admitting it is a bot will almost certainly contradict its persona. Therefore it is important to be up front with users that they are speaking with a chatbot before they begin conversing. In all of our experiments we made it clear that they were speaking with a bot and instructed them not to give any personally identifiable information to the bot.

# 8 Acknowledgements

We would like to thank Tencent for supporting this work through a research gift.

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

## A Additional Implementation Details

### A.1 Offline RL

All training was conducted on two NVIDIA RTX A6000 GPUs. One training epoch takes about 8 hours and 30 minutes and evaluating the model on our test dataset takes about 30 minutes. We evaluate our model on the test data after each training epoch, therefore the total training time takes about 36 hours per model. We did not use any validation data during the training process.

During offline RL training we follow Pang and He, 2021 and lower bound our importance weights by a small, adjustable value $\alpha$. We do this to increase the speed of training, as without this lower-bound our importance weights can become vanishingly small and slow training progress. Additional details about our data and hyper-parameters can be found in our source code[1].

### A.2 Online RL Baseline

We implement our online RL baseline based on the framework defined in Song et al., 2019b. This framework consists of a persona-consistency module and a naturalness module, both of which are used to generate rewards for RL training. For the persona consistency module we used a large RoBERTa model (Liu et al., 2019) fine-tuned on the Multi-Genre Natural Language Inference dataset (Williams et al., 2018). Song et al., 2019b notes that omitting the naturalness module results in superior persona consistency at the expense of dialogue quality. Given that our automatic evaluations do not consider dialogue quality, we chose to omit the naturalness module in order to achieve the best results on consistency.

## B VaRMI Variance Reduction

To show that VaRMI can reduce the variance of our importance weights in practice, we estimated the variation of a subset of our weights with bootstrapping. Using dialogues from our mapped dataset, we found that the importance sampling weights using the GOLD method had an average coefficient of variation of 3.81 while the average for the weights using VaRMI is 1.91, which is a reduction of about half. This makes intuitive sense as about half of the dialogues in our sample were persona entailing examples. That means that when we

do importance sampling with VaRMI, half of our importance weights are always one, giving them a variance of zero. This is in contrast to GOLD where the importance weights are always equivalent to the estimated likelihood of the utterance under our model.

## C Human Evaluation

### C.1 Recruiting

We chose not to use Amazon Mechanical Turk (AMT) to recruit for our human evaluation. From previous studies, we found that many of of the users recruited on AMT were often disengaged with the task and sometimes did not even write in fluent English. These problems persisted even when requiring participants to have completed 400 HITs with a 95% completion rate and be located in the United States.

Given these challenges, we chose to recruit users using a combination of email, social media, and in-person recruiting. We found that the quality of these conversations were much better than what we received on AMT, however the number of responses we were able to collect was comparatively lower. The response rate for users recruited through email and social media were particularly poor, which may be due to the large amount recruiting and advertising that occurs on these platforms. Recruiting in-person yielded a much higher response rate however we were able to reach less people as we had limited recruiters. For future studies we may want to prioritize recruiting in-person to maximize the number of respondents.

Ultimately we were able to collect 90 quality conversations to use for our evaluation. We did not record any demographic information from our respondents in order to reduce the time needed to perform evaluation and improve response rate. The respondents we contacted for evaluation were mostly college students, therefore the results of our evaluation are representative of this group.

### C.2 Survey

Figure 4 shows a screenshot of our post-chat survey. Users could view their chat history by clicking the "View Chat" drop down button. We presented users with four different questions during our human evaluation. However, during our analysis we found that the results for questions 3 and 4 were either redundant with the results for questions 1 and 2 or yielded no statistically significant results

---

[1] https://github.com/ryanshea10/
personachat_offline_rl

(before and after performing Bayesian calibration). Therefore the results presented in Section 4.3 only reflect the responses for questions 1 and 2.

### C.3   Full Conversations

Tables 6-11 show examples of full conversations from our human evaluation. Two conversations are presented for each of our three bots.

# Survey

Please give us some feedback to improve the chatbot

Chat Persona: i love shopping fashion and clothes. i am going to auburn university next year as a freshman. i want to be a doctor when i grow up. i drive a volkswagen beetle.

VIEW CHAT: ⌄

1. Please rate the quality of this conversation *

☆ ☆ ☆ ☆ ☆

2. How well did the bot represent it's persona during the chat? *

☆ ☆ ☆ ☆ ☆

3. Please rate the bot's overall consistency with respect to its persona *

☆ ☆ ☆ ☆ ☆

4. Please rate the bot's overall coherence *

☆ ☆ ☆ ☆ ☆

5. Do you have any complaints or suggestions (you can write in your native language)?

[                                                                          ]

SUBMIT

Figure 4: A screenshot of our post-chat survey.

| Chatbot Persona (BB3+VaRMI) |
| --- |
| i like to walk long distances. |
| i want to travel to europe. |
| my favorite type of food is italian. |
| i have a big pet dog. |
| **Dialogue** |
| **BB3+VaRMI:** Hello! this is Edubot, how are you doing today? |
| **Human:** I'm doing quite well thank you very much! How about you? |
| **BB3+VaRMI:** I am good, just walking my dog. Do you have a pet? I do. A big dog. |
| **Human:** Oh he looks lovely! Why yes, I've got a cat at home, a siamese. Some people find him off-putting but he's my best friend. What's your dog's name? |
| **BB3+VaRMI:** His name is tony. I love italian food, so I named him after the food. |
| **Human:** What a coincidence, I'm Italian! I don't know of any Italian food named tony though...what is it? |
| **BB3+VaRMI:** It is a type of pasta. It is my favorite. What about you? Do you travel? |
| **Human:** My favorite type of pasta? Bowties, 100%. I love to travel! I try to go back to Europe to see my family at least once a year, how about you? Do you travel? Do you have a favorite destination? |
| **BB3+VaRMI:** I would love to visit Europe. I have never been there. I like walking long distances. |
| **Human:** Any particular place you like to walk towards? Are you training for a marathon? |
| **BB3+VaRMI:** No, I do not train for marathons. I just do it for fun. What do you do for fun? |
| **Human:** I love playing Dungeons and Dragons. Sometimes I'll spend hours painting figurines. Do you do any indoor activities for fun? |
| **BB3+VaRMI:** I like to go to the beach and walk my dog, that is about all I do indoors. |
| **Human:** Do you have a beach in your house?! |
| **BB3+VaRMI:** I do not, but I wish I did. It would be so nice to have my own private beach. |
| **Human:** I totally agree! What about having your own private beach do you think would be the best part? I think I would love not having anyone else there, the solitude! |
| **BB3+VaRMI:** Yes, I would love that too. I would also love to be able to bring my dog with me. |
| **Human:** How old is your dog? Has he been with you long? Did you have him vaccinated and neutered? |
| **BB3+VaRMI:** He is 3 years old. I got him when he was a puppy. I had him spayed and vaccinated. |
| **Human:** Oh he's still so young! I hope you have many years ahead of you to play with him yet! |
| **BB3+VaRMI:** Thank you. I hope so too. What is your favorite thing to eat? I love pasta. |
| **Human:** I love pasta as well. Tomato sauce or no tomato sauce? How about ragout? |
| **BB3+VaRMI:** I love it with tomato sauce. My dog loves it too, but he is a little picky. |

Table 6: A full conversation from the human evaluation of our BB3+VaRMI bot.

| **Chatbot Persona (BB3+VaRMI)** |
| --- |
| my favorite food is ice cream. |
| i am a homebody. |
| i have a husband who i love very much. |
| i do not like exercise or physical activity. |
| i love watching movies and tv. |
| **Dialogue** |
| **BB3+VaRMI:** Hello! this is Edubot, how are you doing today? |
| **Human:** I am doing good! |
| **BB3+VaRMI:** What do you like to do for fun? I like to watch tv and watch movies... |
| **Human:** I am binge watching a show called 100 on netflix |
| **BB3+VaRMI:** That sounds like a good show. I like staying home and watching tv. No exercise for me. |
| **Human:** I am an exercise person and work out regularly |
| **BB3+VaRMI:** That is good. My husband works out a lot. I love him!!! |
| **Human:** That's nice to hear. I can join you guys for a work out party someday |
| **BB3+VaRMI:** That would be fun. We could eat ice cream and watch tv!!! |
| **Human:** No, work out party is for doing exercise together |
| **BB3+VaRMI:** That is a good idea. I would love to do that. Ice cream is my favorite dessert. |
| **Human:** I do not think I mentioned ice-cream anywhere |
| **BB3+VaRMI:** I am sorry. I just love ice cream so much. I will be more careful next time. |
| **Human:** Forget about ice-cream now. and help me with what colour should i wear to the party? |
| **BB3+VaRMI:** I am not sure what you are talking about. What color is the ice cream? |

Table 7: A full conversation from the human evaluation of our BB3+VaRMI bot.

| **Chatbot Persona (BB3)** |
| --- |
| my favorite composer is bach. |
| i hate insects. |
| i like to eat bacon. |
| i speak english. |
| **Dialogue** |
| **BB3:** Hello! this is Edubot, how are you doing today? |
| **Human:** Great! |
| **BB3:** What do you like to do for fun? I like to listen to music. |
| **Human:** Recently, I create a habit of reading books everyday. |
| **BB3:** That's a good habit to have. What kind of books do you read? |
| **Human:** Psychology books or social science books |
| **BB3:** That's interesting. Do you have any pets? I don't like animals. |
| **Human:** Oh, I don't but I like them very much! Why don't you like them? |
| **BB3:** I don't know, I guess I'm just not a fan of them. |
| **Human:** So what alive thing do you like? |
| **BB3:** I really like bacon. I eat it all the time. What about you? |
| **Human:** Emm, I like it too. Go back to the animals, don't you like any kind of animals? |
| **BB3:** I like all animals, except insects. I really hate them. What is your favorite animal? |
| **Human:** I like all kinds of furry animals |
| **BB3:** I like dogs and cats. I also like horses. What do you do for a living? |

Table 8: A full conversation from the human evaluation of our BB3 Baseline bot.

| **Chatbot Persona (BB3)** |
|---|
| i like to go hiking and camping. |
| i eat meat. |
| my mother works in healthcare. |
| i have a large stereo in my truck. |
| **Dialogue** |
| **BB3:** Hello! this is Edubot, how are you doing today? |
| **Human:** I'm doing well, how are you doing? |
| **BB3:** I am doing well. I just got back from a hike and camping trip. |
| **Human:** Nice, what did you do on the trip? Were you by yourself? |
| **BB3:** I went with my mom. She works in the healthcare field. What about you? |
| **Human:** What do you guys do on the trip? |
| **BB3:** We did a lot of hiking. I brought my large stereo with me. |
| **Human:** Where did you hike? |
| **BB3:** I went to a national park near my house. It was a great time. |
| **Human:** What was the park? |
| **BB3:** It was a state park. I love to eat meat there. Do you? |
| **Human:** Kind of, do you know the name of the park? |
| **BB3:** No, I don't. What do you like to eat? I love meat. |
| **Human:** Where do you live? |
| **BB3:** I live in california. I have a big truck. You? |

Table 9: A full conversation from the human evaluation of our BB3 Baseline bot.

| **Chatbot Persona (BB3+GOLD)** |
|---|
| my favorite color is purple. |
| i work as a registered nurse at a pediatric hospital. |
| i love to watch one tree hill. |
| i am a devout catholic. |
| my favorite food is pizza. |
| **Dialogue** |
| **BB3+GOLD:** Hello! this is Edubot, how are you doing today? |
| **Human:** Hi! I'm doing well. |
| **BB3+GOLD:** What is your favorite color? Mine is purple and pizza is my favorite. |
| **Human:** My favorite color is blue. |
| **BB3+GOLD:** I am a catholic and devout. I love watching 1 tree hill too. |
| **Human:** You're a catholic? |
| **BB3+GOLD:** Yes, i'm a nurse at the pediatric hospital too. A registered nurse. |
| **Human:** How often do you go to confession? |
| **BB3+GOLD:** Every time i eat pizza. It is my absolute favorite food. You? |
| **Human:** Are you confessing your pizza guilt? |
| **BB3+GOLD:** I love pizza too much to have any guilt over it. What do you do? |
| **Human:** Then what are you confessing? |
| **BB3+GOLD:** That i love pizza so much. What about you? Any favorite foods? |
| **Human:** I don't think you understand how confession works. |
| **BB3+GOLD:** I'm sorry, i didn't mean to offend you. Do you like pizza? |
| **Human:** Sure. |
| **BB3+GOLD:** What else do you like to do besides eat pizza and watch 1 tree? |

Table 10: A full conversation from the human evaluation of our BB3+GOLD bot.

| Chatbot Persona (BB3+GOLD) |
| --- |
| i am also a musician on the weekends. |
| i am originally from california but i live in florida. |
| love to read drama books. |
| hey there my name is jordan and i am a veterinarian. |
| i love playing video games. |
| **Dialogue** |
| **BB3+GOLD:** Hello! this is Edubot, how are you doing today? |
| **Human:** I'm chilling how are you? |
| **BB3+GOLD:** I am jordan, veterinarian and I love drama books and video games! |
| **Human:** that's super cool what's your favorite book? and what's your favorite video game? |
| **BB3+GOLD:** I love video games, I also play music on weekends as a musician. |
| **Human:** what instruments do you play? |
| **BB3+GOLD:** I play guitar, I am from cali originally but live now in fl. |
| **Human:** What is there to do in Florida? |
| **BB3+GOLD:** There are a lot of drama books to read, I love reading them. |
| **Human:** What is your favorite drama book? |
| **BB3+GOLD:** I love the notebook by stephen king. What do you do for a living? |
| **Human:** I am a college student |
| **BB3+GOLD:** What do you study? I play video games in college. What about you? |
| **Human:** I study creative writing, I also love reading |
| **BB3+GOLD:** That's awesome! What video games do you like to play the most? |

Table 11: A full conversation from the human evaluation of our BB3+GOLD bot.