# OpenReview forum: "Building Persona Consistent Dialogue Agents with Offline Reinforcement Learning"
_EMNLP/2023/Conference — EMNLP 2023 Main_

### Official Review · Reviewer_vjbp · 2023-08-04

**Soundness:** 3

**Excitement:**

2: Mediocre: This paper makes marginal contributions (vs non-contemporaneous work), so I would rather not see it in the conference.

**Paper Topic And Main Contributions:**

This paper is about building persona consistent dialogue agents. A persona describing attributes of the chatbot is provided such as  "I have a cat, I am blonde, I am a musician”. During the conversation, the chatbot is required adhere this persona, i.e. be consistent. Instead of relying on supervised or online reinforcement learning (RL), the authors propose the usage of offline RL in order to 1) train the model without the need for generating more data and 2) punishing and rewarding specific utterances using RL. The authors use policy-gradient RL for optimising their model, which requires importance sampling as the data has been collected by a different policy. As importance sampling can result in high variance, the authors propose the “VaRMI Importance Sampling” to reduce variance. The proposal is based on heuristics where the importance weight is ignored completely for positive reward candidates and the denominator of the importance weight is ignored for negative reward candidates. The method is thus only applicable if there is "some notion" of positive and negative reward for the task.

**Questions For The Authors:**

I highly appreciate answers to the questions stated in the “reasons for rejection”.

**Reasons To Accept:**

The usage of offline RL is a thriving field and worth investigating for the task of persona-consistent dialogue.

**Reasons To Reject:**

I have concerns about the importance sampling assumptions. I don’t see the validity of the assumption that the likelihood of pi_b for all training samples (or the ones with negative reward candidates) should be the same. This sounds like a very strong assumption to me. It could be validated empirically by evaluating the probability of the MLE policy.
Similarly, I don’t see why the importance weight can be assumed to be 1 for positive reward candidates. This might hold at the beginning of training, but the offline RL optimization can push pi away from pi_b. The authors justify that statement by saying that there is a large amount of positive examples in the data already, i.e. assuming a large coverage. This will result in an additional assumption. However, the fact that the policy moves away from the behaviour policy is an issue in offline RL that has led to several behaviour regularization strategies (“Behaviour Regularised Offline RL”, Wu et. al. 2019) and is completely neglected by this work. Or is this assumption valid due to the reward design of +-1? Moreover, if the assumption holds, should it make a difference whether to use pi/mu or not?

The outline of the paper can be improved to facilitate understanding of the reader.

A. How are actions and state defined in your framework? I assume that an action is a word/token and a trajectory is given by a sequence of words.

B. How is hat(Q) (Line 236) calculated in your experiments? Is it given by the critic that is mentioned? Or somehow a Monte-Carlo return? In RL the critic typically estimates the expected return, whereas in this work the critic seems to output a reward (line 212). So is the critic a reward function in your case? (Line 215)

C. The explanation of the offline RL framework (Section 3.3) is very hard to parse. Typically the offline RL setup starts with a dataset D of tuples (s_t, a_t, r_t, s_t+1, terminal). Is (s_t, a_t, s_t+1, terminal) given by PersonaChat and then you infer the corresponding reward r_t from DNLI dataset? Or do you augment PersonaChat somehow with examples from DNLI? How does the resulting dataset look like? Is it different to the dataset used for supervised learning?

D. Line 275: You assume that pi_theta has already been trained on a large amount of positive reward examples. Does that mean that you assume the supervised training data to be an expert dataset? Are there any assumption on the supervised data for your method?

E. How is the loss calculated in Figure 3? Is it based on the Equation in line 202 but with importance sampling weights?

F. The vanilla importance sampling pi/mu should be another baseline to showcase the actual need for variance reduction. In addition, what is the performance if you omit the importance weight completely (and not only for positive examples)?

G. It is confusing that the framework is explained in the section “Implementation”

H. Variance reduction for importance weights emerged first in off-policy learning. It might be worth mentioned related work about it, such as (Safe and Efficient Off-Policy RL, Munos et. al. 2016).

I. Line 288: What does “There is some notion of a positive and negative reward” mean? Is it enough that there is  both positive and negative reward or does it have to be +-1 only?

**Reproducibility:**

3: Could reproduce the results with some difficulty. The settings of parameters are underspecified or subjectively determined; the training/evaluation data are not widely available.

**Reviewer Confidence:**

4: Quite sure. I tried to check the important points carefully. It's unlikely, though conceivable, that I missed something that should affect my ratings.

---

> ### Author Rebuttal · Authors · 2023-08-28
>
> Thank you for your detailed comments and insightful questions. Below are our responses to your questions and concerns.
>
> *I don’t see the validity of the assumption that the likelihood of pi_b for all training samples (or the ones with negative reward candidates) should be the same. This sounds like a very strong assumption to me.*
>
> This assumption is taken from the importance sampling scheme proposed in the GOLD algorithm (Pang and He, 2019). To quote from the paper "Another obstacle is that we do not know $\pi_b$ which produced the demonstrations $D = \{(x^{(i)} , y^{(i)} )\}^N_{i=1}$. One option is to estimate $\pi_b$ on $D$. Here we take a simpler approach that uses the empirical distribution: $\pi_b(\tau) ≈ 1/N$ for $\tau \in D$ and 0 otherwise." Empirically this assumption has been shown to work well in the original paper as well as others that have used the GOLD algorithm as a baseline (Li et al., 2022; Snell et al. 2023). Based on your comments, we attempted to validate this assumption by looking at the distribution of $\tau$ in $D$. We found that there were no duplicates and that they follow a uniform distribution under our dataset. We also have no reason to assume that any expert demonstration in our dataset is more valid than any of the other demonstrations and therefore should not have a different likelihood. Given this, we feel as though this assumption is valid and well-supported by prior work. We hope this helps alleviate your concern, we will add additional details to sections 3.1 and 3.2 to better explain this justification.
>
> *I don’t see why the importance weight can be assumed to be 1 for positive reward candidates. This might hold at the beginning of training, but the offline RL optimization can push pi away from pi_b. The authors justify that statement by saying that there is a large amount of positive examples in the data already, i.e. assuming a large coverage. This will result in an additional assumption. However, the fact that the policy moves away from the behaviour policy is an issue in offline RL that has led to several behaviour regularization strategies (“Behaviour Regularised Offline RL”, Wu et. al. 2019) and is completely neglected by this work. Or is this assumption valid due to the reward design of +-1? Moreover, if the assumption holds, should it make a difference whether to use pi/mu or not?*
>
> We assume that the importance weight is 1 for our positive reward examples because our dialogue policy has already been trained on persona entailing dialogues using imitation learning. Therefore we assume that our dialogue policy, $\pi_\theta$, has learned the behavioral policy, $\pi_b$, for generating positive examples to an arbitrary degree which makes $\frac{\pi_\theta}{\pi_b} \approx 1$ for positive reward examples. This works for our case due to the reward design of +-1 and may work for other cases where there are absolute positive and negative rewards (see our response to I. for more details on this). The difference between this and the standard method of importance sampling would be the reduction of variance that comes due to the elimination of these weights.
>
> As for your concern about $\pi_\theta$ diverging from $\pi_b$ during training, we feel that we didn't encounter this issue for the following reasons. Prior work has suggested that this divergence issue arises in settings where value function approximation is needed to obtain Q-values (Van Hasselt et al., 2018; Wu et. al., 2019). In our case we are able to train on ground truth rewards that come from human annotations, which eliminates the need for value function approximation. Prior work has also shown that when this issue arises the learned policy performs poorly, especially when attempting to generalize beyond the dataset distribution (Fujimoto et al., 2018; Kumar et al., 2019). Given that we observe positive results during our human and automatic evaluations we believe that we did not encounter any policy divergence issues during training. Due to these reasons, we did not explore any behavior regularization methods. We will add more details to section 3.2 to better clarify this and justify why we did not explore behavior regularization.
>
> *The outline of the paper can be improved to facilitate understanding of the reader.*
>
> Thank you for pointing this out, based on the comments we have received the main source of confusion seems to arise from a lack of details regarding our importance sampling methods. We add additional details to clarify this in section 3. If you have any other specific points of criticism regarding the structure please let us know so we can correct the issues.
>
> *A. How are actions and state defined in your framework? I assume that an action is a word/token and a trajectory is given by a sequence of words.*
>
> Your assumption about the actions and trajectories is correct. Our state is defined as the current persona and dialogue context, which we concatenate together.
>
> *B. How is hat(Q) (Line 236) calculated in your experiments? Is it given by the critic that is mentioned? Or somehow a Monte-Carlo return? In RL the critic typically estimates the expected return, whereas in this work the critic seems to output a reward (line 212). So is the critic a reward function in your case? (Line 215)*
>
> In our setting, $\hat{Q}$ comes from human annotations originating from the DNLI dataset. These annotations are treated as ground truth, sequence level rewards which map directly to the items in our offline dataset (more details on the dataset in our response to C). This mapping is generated during the process we use to create our offline dataset. So our critic is essentially a fixed mapping between our offline samples and the human annotated rewards. This mapping is then used to retrieve rewards during training. During inference we do not calculate any Q-values and simply run our experiments using the modified dialogue policy.
>
> *C. The explanation of the offline RL framework (Section 3.3) is very hard to parse. Typically the offline RL setup starts with a dataset D of tuples (s_t, a_t, r_t, s_t+1, terminal). Is (s_t, a_t, s_t+1, terminal) given by PersonaChat and then you infer the corresponding reward r_t from DNLI dataset? Or do you augment PersonaChat somehow with examples from DNLI? How does the resulting dataset look like? Is it different to the dataset used for supervised learning?*
>
> Our dataset looks more like $(s_t, \tau, r_t)$ where $\tau$ is a sequence of words (a dialogue utterance). This is because we only use one sequence level reward and can infer intermediate and terminal steps when needed from the trajectory. Our dataset is created by augmenting the PersonaChat dataset with the DNLI data along with using the DNLI dataset to infer rewards. Each datapoint in the DNLI dataset contains statement pairs along with entailment labels (which we use as rewards) like: (sentence1, sentence2, label), while the PersonaChat dataset just contains persona sets and dialogues. The sentences in the DNLI dataset are either personas from the PersonaChat dataset or dialogue utterances from the PersonaChat dataset. To perform our mapping, we scan the DNLI dataset for cases where one sentence maps to a dialogue utterance and the other maps to a persona. We then add this persona to the example in the PersonaChat dataset (filtering out any existing personas in the set that contradict the one we inserted) and use the dialogue utterance as $\tau$ and the label as the reward. Below is an example to illustrate how this mapping is done, as well as what a final datapoint would look like (table 1 also shows two examples).
>
>
> | DNLI      | PersonaChat | Mapped |
> | ----------- | ----------- |  ----------- |
> | ("I like milk", "No, I don't like dairy", -1)| your_persona: I'm lactose intolerant| your_persona: I like milk |
> ||your_persona: I am female|your_persona: I am female|
> ||P1: Hi, do you like cereal?|P1: Hi, do you like cereal?|
> ||P2: No, I don't like dairy|P2: No, I don't like dairy         |
> |||reward: -1         |
>
> In this case "No, I don’t eat dairy" would be used as $\tau$ and everything preceding it would be used as the state. This data is very similar to supervised learning with the addition of the reward that we have associated with our targets.
>
> *D. Line 275: You assume that pi_theta has already been trained on a large amount of positive reward examples. Does that mean that you assume the supervised training data to be an expert dataset? Are there any assumption on the supervised data for your method?*
>
> Yes, we assume that the supervised training dataset is an expert dataset containing positive examples. This is the only assumption we make. In our case this assumption is satisfied by the fact that BB3 has been fine-tuned on PersonaChat and achieves a perplexity of around 5.8 (Shuster et al., 2022).
>
> *E. How is the loss calculated in Figure 3? Is it based on the Equation in line 202 but with importance sampling weights?*
>
> This loss is simply vanilla NLL loss (no importance sampling). We will update the figure to make this clear.
>
> *F. The vanilla importance sampling pi/mu should be another baseline to showcase the actual need for variance reduction. In addition, what is the performance if you omit the importance weight completely (and not only for positive examples)?*
>
> Would you mind providing some additional clarification on this additional baseline you're suggesting? Is there another assumption you suggest we make about our behavioral policy? We would be happy to add an additional importance sampling baseline. So far we have considered the method proposed in GOLD to be our vanilla/baseline method due to the reasons described above, as well as the fact that, to our knowledge, it is the only importance sampling method that has been successfully applied to text generation in cases where the behavioral policy is unknown.
>
> If we eliminate the importance weight completely, the loss on both our positive and negative examples skyrockets. In fact, when we tried this we were unable to complete training as we kept encountering overflows (although we may be able to get around this by using an extremely small learning rate, we tested as low as 1e-8). We believe this is due to the issue of policy divergence. We did not mention this in the paper since we had no metrics to report and the issue was easily resolved by including importance weights. However, we will add a section about this in the appendix if you feel it would be worthwhile.
>
> *G. It is confusing that the framework is explained in the section “Implementation”*
>
> Thank you for pointing this out, we will split the "Implementation" subsection into two subsections to improve the structure.
>
> *H. Variance reduction for importance weights emerged first in off-policy learning. It might be worth mentioned related work about it, such as (Safe and Efficient Off-Policy RL, Munos et. al. 2016).*
>
> Thank you for this suggestion, we will include a discussion on this in section 2.
>
> *I. Line 288: What does “There is some notion of a positive and negative reward” mean? Is it enough that there is both positive and negative reward or does it have to be +-1 only?*
>
> This means that the setting we are studying has absolute positive or negative associations to the actions we are performing. For example, in persona consistency we know that entailing your persona is good and contradicting it is bad. This may not be the case in all settings. As a simple example, let's say we are training a robot to throw a ball at a target and use the distance from the target as a reward. In this case we would not have any notion of a positive or negative, just "better" or "worse". We may choose to subtract our rewards by a baseline to reduce variance, which would give us positive and negative rewards. However, we would not be justified in eliminating the importance weights for these examples as we have no notion of absolute positive or negative. There is some theoretical justification for why this method could work in any case with positive and negative rewards and not just +-1. However, more work needs to be done to see if this can achieve positive results in practice. We hope to explore this more in the future.
>
> Thank you again for your detailed review. We hope this response provides clarification to your questions and alleviates your concerns about recommending this paper for acceptance. Please let us know if you have any additional questions or comments.

---

### Official Review · Reviewer_Fcrt · 2023-08-05

**Soundness:** 2

**Excitement:**

3: Ambivalent: It has merits (e.g., it reports state-of-the-art results, the idea is nice), but there are key weaknesses (e.g., it describes incremental work), and it can significantly benefit from another round of revision. However, I won't object to accepting it if my co-reviewers champion it.

**Paper Topic And Main Contributions:**

This paper introduces an approach for constructing persona-consistent dialogue agents using offline reinforcement learning. Furthermore, it presents a straightforward importance sampling technique aimed at reducing the variance of importance weights in offline reinforcement learning scenarios.

**Questions For The Authors:**

Please refer to "Reasons To Reject."

**Reasons To Accept:**

Persona consistency is a crucial topic, and the paper's attempt to address it using offline reinforcement learning is a commendable approach.

**Reasons To Reject:**

a. The paper lacks a clear and coherent structure, making it difficult to follow and understand the presented ideas.

b. The motivation of the paper is contradictory. It addresses the issue of high training costs associated with online RL methods, yet it requires manually annotated dialogues for training, which further need to undergo subsequent selection.

c. The paper states that the proposed method can ensure agent consistency but does not provide an explanation for how this solution is achieved.

d. The necessity for introducing the new important sampling method in the paper is not established, given the existence of many widely applied importance sampling techniques. The authors have not clarified the necessity and advantages of their proposed new method.

e. The paper lacks appropriate baseline comparisons, only contrasting with BB3 and omitting comparisons with methods that address the same issue proposed in the paper. For instance, the drawbacks of multistage re-writing and Bayesian rational speech acts methods are not mentioned by the authors. This omission weakens the substantiation of the paper's contributions.

**Reproducibility:**

4: Could mostly reproduce the results, but there may be some variation because of sample variance or minor variations in their interpretation of the protocol or method.

**Reviewer Confidence:**

3: Pretty sure, but there's a chance I missed something. Although I have a good feel for this area in general, I did not carefully check the paper's details, e.g., the math, experimental design, or novelty.

---

> ### Author Rebuttal · Authors · 2023-08-28
>
> Thank you for your detailed and considered review of our manuscript. Below are our responses to you questions.
>
> *a. The paper lacks a clear and coherent structure, making it difficult to follow and understand the presented ideas.*
>
> Thank you for pointing this out, based on the comments we received there seems to some confusion surrounding the technical details of our offline RL algorithm, specifically our importance sampling. We will expand section 3 along with its subsections to better explain the details behind our framework and offline RL method. If you have any other specific points of criticism regarding the structure, please let know so we can correct this issue.
>
> *b. The motivation of the paper is contradictory. It addresses the issue of high training costs associated with online RL methods, yet it requires manually annotated dialogues for training, which further need to undergo subsequent selection.*
>
> In this paper we make a distinction between training costs (training time, memory usage, etc.) and data collection costs (recruiting annotators, annotation time, etc.). Online RL has a low data collection cost as the model generates its own samples during training, but a high training cost since the process of generating these samples is expensive. Whereas offline RL has these costs reversed. In our setting of persona consistency, there are annotated datasets which have already been collected, which means that offline RL will have both low data collection costs and low training costs. Therefore, our paper is motivated by the the fact that we can use offline RL to leverage existing datasets and improve training efficiency for persona consistency, without the need to collect additional data. Furthermore, our offline RL method allows us to use ground truth rewards during training and avoid zero reward regions which further improves efficiency. Online RL on the other hand requires value function approximation which may result in a noisy reward or a reward which is close to zero, both of which hamper the training process. We will make changes to the intro to make these points more clear.
>
> *c. The paper states that the proposed method can ensure agent consistency but does not provide an explanation for how this solution is achieved.*
>
> Our method achieves persona consistency by explicitly rewarding  persona entailing utterances and punishing contradictory utterances using offline RL. By doing this we make our agent less likely to utter phrases that contradict its persona and more likely to utter phrases that entail it, which improves consistency. This is an improvement over supervised learning where agents are never punished for uttering contradictions and are less consistent as a result. We will make this point clearer in the introduction.
>
> *d. The necessity for introducing the new important sampling method in the paper is not established, given the existence of many widely applied importance sampling techniques. The authors have not clarified the necessity and advantages of their proposed new method.*
>
> We introduce the new importance sampling method as a way to reduce the variance of importance sampling weights. Policy gradient based offline RL methods have seen limited use in practice due to the high variance which comes as a result of importance sampling (Levine et al., 2020). High variance causes the resulting importance weights to become degenerate and adds too much noise to the gradient estimates to allow for effective optimization. While many importance sampling methods have been proposed, to our knowledge the only method that has been successfully applied to text generation in a fully offline setting is the one proposed in GOLD (Pang and He, 2021). The usage of policy-gradient offline RL in dialogue settings is fairly under-explored therefore there are many opportunities to improve upon existing methods. Given this we introduce a method which takes additional steps to reduce importance weight variance by eliminating them for positive reward examples. Therefore the main advantage behind our method is the reduced variance of our importance weights. We will add this clarification to the introduction as well as section 3.2.
>
> *e. The paper lacks appropriate baseline comparisons, only contrasting with BB3 and omitting comparisons with methods that address the same issue proposed in the paper. For instance, the drawbacks of multistage re-writing and Bayesian rational speech acts methods are not mentioned by the authors. This omission weakens the substantiation of the paper's contributions.*
>
> The drawback of multistage rewriting (Song et al., 2020) is the fact that it doesn't handle multi-turn persona consistency. To quote from the paper "We focus on the persona consistency of single-turn responses, and we leave the modeling of multi-turn persona consistency as future work." The drawback of incorporating the Bayesian RSA framework (Kim et al., 2020) is the increased computational cost during inference time due to the modified decoding scheme. This results in longer response times from the bot as well as the need for greedy decoding, which reduces response diversity and dialogue quality. We will mention these drawbacks in section 2. We will also add baselines for Bayesian RSA and online RL (Song et al., 2019) to our automatic evaluation. We have included a modified version of table 2 at the end of the rebuttal which includes the online RL baseline. We will work to include the Bayesian RSA baseline as soon as possible.
>
> Thank you again for your review, we hope these responses provide clarifications to your questions and alleviate your concerns about recommending this paper for acceptance.
>
> Modified version of Table 2
> |Model|Hits@1&uarr;|Entail@1&uarr;|Rand@1&darr;|Contradict@1&darr;|
> |-----| -----------|--------------|------------|------------------|
> | BB3 |26.6        | 29.5         | 13.3       |30.6
> | BB3+RL |27.5   | 30.4         | 10.9        |31.2
> | BB3+GOLD |37.5*   | 37.3*         | 5.4*        |**19.9***
> | BB3+VaRMI|**37.6*** | **37.6***   | **4.4***    |20.3*
> \*denotes statistically significant difference compared to the BB3 baseline (two-sample z-test, $p < 0.05$)

---

### Official Review · Reviewer_2MMu · 2023-08-05

**Soundness:** 4

**Excitement:**

3: Ambivalent: It has merits (e.g., it reports state-of-the-art results, the idea is nice), but there are key weaknesses (e.g., it describes incremental work), and it can significantly benefit from another round of revision. However, I won't object to accepting it if my co-reviewers champion it.

**Paper Topic And Main Contributions:**

The paper's objective is to utilize Offline RL for enforcing consistency of Dialogue Agents with a provided persona. To achieve this goal, the authors introduce Variance-Reducing MLE-Initialized (VarMI) importance sampling, a straightforward policy gradient-based Offline RL algorithm designed to mitigate variance in importance sampling weights of policy gradients through approximations. This method is then applied to fine-tune BlenderBot3 (BB3) using the Dialogue Natural Language Inference (DNLI) dataset and PersonaChat. To evaluate the model, the authors compare against GOLD and baseline BB3. Metrics used encompass exact match, relaxed match, contradictions, and random utterances. The results demonstrate that VaRMI outperforms in all metrics except for contradictions, where GOLD exhibits superior performance. Human evaluations are also conducted, show that VaRMI achieves higher dialogue quality, while GOLD maintains greater persona consistency.

**Questions For The Authors:**

Q1. On line 279-280 you make approximations for importance sampling weights. Would it be possible to show that these reduce the variance of gradients in practice?

Q2. You don't compare against any other Offline RL methods. Would it be possible to compare against others to see whether the increase in performance is caused by VaRMI or RL training in general?

Q3. In Table 3 you list which results are statistically significant. Is it possible to calculate this for Table 2 as well? Some of the numbers seem close, ex. 37.5 vs 37.6 for Hits@1

**Reasons To Accept:**

S1. **Applicability**: Personalization is an important industry topic, so this work will be widely applicable

S2. **Reproducibility**: Authors provide code to reproduce results easily.

**Reasons To Reject:**

R1. **Weak Results**: The results are a bit weak; human evaluations show that GOLD is more persona consistent than VarMI. Additional results from the authors during the rebuttal indicates that results between VarMI and GOLD are not statistically significant, despite a decrease in the variance of importance sampling weights.

**Reproducibility:**

5: Could easily reproduce the results.

**Reviewer Confidence:**

4: Quite sure. I tried to check the important points carefully. It's unlikely, though conceivable, that I missed something that should affect my ratings.

---

> ### Author Rebuttal · Authors · 2023-08-28
>
> Thank you for your helpful and constructive comments. Before answering your questions, we would like to make one small clarification based on your summary and comments in R1.
>
> Both VaRMI and GOLD refer to policy gradient offline RL methods, the difference between them is based on the type of importance sampling we use. GOLD uses the importance sampling scheme defined in the GOLD algorithm presented in Pang and He, 2021. VaRMI follows the GOLD method of importance sampling but with some additional assumptions to reduce the variance of the importance weights. We use the base BB3 model as our supervised fine-tuned baseline. BB3 has already been fine-tuned on the PersonaChat dataset and achieves a perplexity of around 5.8 (Shuster et al., 2022). The authors of BB3 note that performing additional fine-tuning on the training datasets results in significant overfitting (Shuster et al., 2022). We observed this overfitting as well when we performed additional fine-tuning on PersonaChat and therefore felt it was best to use the base version of BB3 as our baseline. We will add some additional comments about this in sections 3.1 and 3.3 in order make this more clear.
>
> *Q1. On line 279-280 you make approximations for importance sampling weights. Would it be possible to show that these reduce the variance of gradients in practice?*
>
> Thank you for this suggestion. We modified your proposed experiment slightly to measure the reduction in variance of our importance sampling weights directly since that is the intention of those approximations. While reducing the variance of our importance weights should also reduce the variance of our gradients, this can be influenced by other factors such as batch size. Therefore we felt it would be better to look the variance of the importance weights directly. We did this by bootstrapping a random sample of our importance weights using both the GOLD and VaRMI importance sampling methods. Our importance weights have some fairly significant variations in magnitude, therefore we chose to use a standardized measure of dispersion called the coefficient of variation when aggregating the sample. The importance sampling weights using the GOLD method have an average coefficient of variation of 3.81 while the average for the weights using VaRMI is 1.91, which is a reduction of about half. We will include this result in the appendix.
>
> *Q2. You don't compare against any other Offline RL methods. Would it be possible to compare against others to see whether the increase in performance is caused by VaRMI or RL training in general?*
>
> We did not include any other offline RL algorithms in our automatic evaluations due to the fact that, to our knowledge, existing offline RL algorithms that have been applied to a dialogue setting (aside from GOLD) use Q-learning (Verma et al., 2022; Jaques et al., 2020; Snell et al., 2023). Our automatic evaluations are designed to evaluate the dialogue policy directly, which works for policy gradient or actor-critic algorithms that actively update the dialogue policy. Since Q-learning algorithms do not modify the dialogue policy we cannot apply them to our automatic evaluations. However, given the comments we have received about our lack of baselines, we will add additional baselines to our automatic evaluations. One which modifies BB3 with policy gradient online RL training (Song et al., 2019) as well as one which imbues BB3 with pragmatic self-consciousness using the Bayesian Rational Speech Acts framework (Kim et al., 2020). We have included a modified version of table 2 at the end of the rebuttal which includes the online RL baseline. We will work to include the Bayesian RSA baseline as soon as possible.
>
> *Q3. In Table 3 you list which results are statistically significant. Is it possible to calculate this for Table 2 as well? Some of the numbers seem close, ex. 37.5 vs 37.6 for Hits@1*
>
> Yes, thank you for the suggestion. Based on two-sample z tests we find that all of the differences between both of our offline RL methods and the BB3 baseline are statistically significant. We also find that none of the differences between the GOLD and VaRMI offline RL methods are statistically significant. We will add these results to Table 2.
>
> We hope this clarifies your questions. If you have any additional questions or concerns please let us know.
>
> Modified version of Table 2
> |Model|Hits@1&uarr;|Entail@1&uarr;|Rand@1&darr;|Contradict@1&darr;|
> |-----| -----------|--------------|------------|------------------|
> | BB3 |26.6        | 29.5         | 13.3       |30.6
> | BB3+RL |27.5   | 30.4         | 10.9        |31.2
> | BB3+GOLD |37.5*   | 37.3*         | 5.4*        |**19.9***
> | BB3+VaRMI|**37.6*** | **37.6***   | **4.4***    |20.3*
> \*denotes statistically significant difference compared to the BB3 baseline (two-sample z-test, $p < 0.05$)

---

### Meta-Review · Area_Chair_k2ko · 2023-09-17

**Recommendation:** 3

**Metareview:**

This paper presents a novel offline RL method for training chatbots with consistent persona.  Experiment results show that the method outperforms the BlenderBot3 baseline across several metrics, and is comparable to an alternative offline RL method.  The reviewers consider this an important topic and find the proposed method interesting, but also point out the the clarity of the paper could be improved.  The authors have provided extensive rebuttals to the reviews and the reviewers have acknowledged this.  Overall, the work looks sound and interesting, though the reviewers still have some reservations about it.

---

### Meta-Review · Senior_Area_Chairs · 2023-10-05

**Recommendation:** 3

**Metareview:**

meta review

---

### Decision · Program_Chairs · 2023-10-07

**Decision:**

Accept-Main

**Comment:**

This paper presents a novel offline RL method for training chatbots with consistent persona.  Experiment results show that the method outperforms the BlenderBot3 baseline across several metrics, and is comparable to an alternative offline RL method.  The reviewers consider this an important topic and find the proposed method interesting, but also point out the the clarity of the paper could be improved.  The authors have provided extensive rebuttals to the reviews and the reviewers have acknowledged this.  Overall, the work looks sound and interesting, though the reviewers still have some reservations about it.|meta review